# The 'wickedness' of governing land subsidence: Policy perspectives from urban Southeast Asia

Rapti Siriwardane-de Zoysa[1]*, Tilo Schöne[2], Johannes Herbeck[3], Julia Illigner[4], Mahmud Haghighi[5], Hendricus Simarmata[6], Emma Porio[7], Alessio Rovere[8], Anna-Katharina Hornidge[9]

1 Department for Social Sciences, Leibniz Centre for Tropical Marine Research (ZMT), Bremen, Germany, 2 Geodetic Hazard Monitoring Group, Section Global Geomonitoring and Gravity Field, GFZ German Research Centre for Geosciences, Potsdam, Germany, 3 Artec Sustainability Research Centre, University of Bremen, Bremen, Germany, 4 Geodetic Hazard Monitoring Group, Section Global Geomonitoring and Gravity Field, GFZ German Research Centre for Geosciences, Potsdam, Germany, 5 Helmholtz Centre Potsdam–GFZ German Research Centre for Geosciences, Potsdam, German, 6 Centre for Urban and Regional Studies, Universitas Indonesia, Jakarta, Indonesia, 7 Ateneo de Manila University; and Manila Observatory, Quezon City, Metro Manila, Philippines, 8 Centre for Marine Environmental Sciences (MARUM), University of Bremen, Bremen, Germany, 9 Institute for Sociology, University of Bremen; and Leibniz Centre for Tropical Marine Research (ZMT), Bremen, Germany

☯ These authors contributed equally to this work.
* rapti.siriwardane@leibniz-zmt.de

**Data Availability Statement:** Due to ethical and legal restrictions, anonymized aggregated

## Abstract

Drawing on Jakarta, Metro Manila and Singapore as case studies, we explore the paradox of slow political action in addressing subsiding land, particularly along high-density urban coastlines with empirical insights from coastal geography, geodesy analysis, geology, and urban planning. In framing land subsidence as a classic 'wicked' policy problem, and also as a hybrid geological and anthropogenic phenomenon that is unevenly experienced across urban contexts, the paper uses a three-step analysis. First, satellite-derived InSAR maps are integrated with Sentinel-1A data in order to reveal the socio-temporal variability of subsidence rates which in turn pose challenges in uniformly applying regulatory action. Second, a multi-sectoral mapping of diverse policies and practices spanning urban water supply, groundwater extraction, land use zoning, building codes, tenurial security, and land reclamation reveal the extent to which the broader coastal governance landscape remains fragmented and incongruous, particularly in arresting a multi-dimensional phenomenon such as subsidence. Finally, in reference to distinct coastal identities of each city–the 'Sinking Capital' (Jakarta), 'Fortress Singapore', and the 'Disaster Capital' (Manila) the paper illustrates how land subsidence is portrayed across the three metropolises in markedly similar ways: as a reversible, quasi-natural, and/or a highly individualized problem.

## 1. Introduction

Global sea level rise is one of the most direct consequences of the changing climate and is the result of two main processes [1]. First, eustatic changes are driven by ice melting, steric effects

qualitative interview data can be made available upon request from qualified, interested researchers. The following non-author institutional contact who will be available to field future data requests is: Prof. Dr. Nils Moosdorf Head of Biogeochemistry and Geology, Leibniz Centre for Tropical Marine Research (ZMT) Email: nils. moosdorf@leibniz-zmt.de phone: +49 421 23800 – 33 Meanwhile, we have attached a minimal (aggregated) data set for the social survey results. The qualitative interviews (as entire monologue transcripts) cannot be shared as this data is sensitive and much of the content potential reveals participant identities. A prerequisite in soliciting verbal informed consent was that the identities of the participants would be kept confidential and that no access to raw data would be given to members outside the research team.

**Funding:** This study was generously funded by the Deutsche Forschungsgemeinschaft / the German Science Foundation (DFG), through the first phase of the Special Priority Program (SPP) - 1889 "Regional Sea Level Change and Society." The paper integrates interdisciplinary work of three SPP 1889 research projects – "Epistemic Mobilities and the Governance of Environmental Risks in Island Southeast Asia" (EMERSA), the "Hazardous Potential in Indonesia and South East Asia" (CoRSEA), and "Holocene Sea-level Changes in Southeast Asia" (SEASchange).

**Competing interests:** The authors report no conflict of interest in this research paper.

and the redistribution of continental and sea surface water. Second, vertical land movements cause relative changes in the sea level. At many locations globally, the impacts of eustatic sea level rise are exacerbated by coastal subsidence [2, 3]. For this reason, the subsidence of coastal areas has been a subject of research since the mid-twentieth century [4, 5]. Most urban sites affected by land subsidence (LS) are located in delta and peat-heavy regions. Examples include Bangkok, Jakarta, Metro Manila, Ho Chi Minh City, Shanghai, New Orleans, or Venice) [4, 6, 7], including hinterland regions such as Central Mexico [8], the Eastern Beijing Plain [9], and California's Central Valley [10].

Conventionally, most policy has emphasised the global and regional projections of mean sea level rise, yet land subsidence rates have presented more pressing and localised challenges. Land subsidence (LS), as a socio-ecological problem, has therefore generated less policy traction at the municipal level. Here, intensified narratives and images of the 'sinking city' only began to appear in international and local media accounts within the last decade and a half.

Steadily increasing policy and public recognition, particularly in coastal deltas and peatlands prone to sea level rise (SLR), appears to be a more recent trend [11, 12], against more visibly articulated phenomena such as ice sheet melt, coastal erosion, and the interactional effects of El Niño and La Niña [13, 14]. Furthermore, the Intergovernmental Panel on Climate Change (IPCC) formulation of the Representative Conservation Pathway 2.6 to keep global warming at 2˚C [15: p. 26]–which became especially salient in the climate negotiations in Paris in 2015 –continues to guide attention to globally induced SLR and the need for *global* action [16]. This overemphasis arguably diffuses the urgency of national and municipally action for governing land subsidence at local level.

Most public accounts in media discourses and in public engagement projects that combine photographic and other forms of visual art have been relatively uniform in their storying of the sinking city: 'Jakarta is sinking so fast, it could end up underwater', relays a—*New York Times* article that is awash with hauntingly dystopian images of crumbling flood walls and floating neighbourhoods [17]. Although the anthropogenic effects that lead to land subsidence (LS) are often described in their narrative accounts, its *visual* appearance seems to be more dramatically represented only at the surface level. This paper therefore aims at exploring the distance between discourses on catastrophe and urgency, against apparent local policy inaction (or non-action) in three distinct Southeast Asia capitals–Jakarta (Indonesia), Metro Manila (the Philippines), and the island city-state of Singapore. We begin by considering the complexities in perceiving LS as a multidimensional, socio-ecological phenomenon, and as one of the world's most "underrated problems" [18].

LS entails a decrease in surface elevation because of a gradual settling or sudden movement of subsurface support material [19]. Studies on the causes of LS have focused on the overexploitation of groundwater aquifers, while the compounding effects of sediment compaction on LS have barely been explored in policy-focused studies, particularly those providing inputs for building codes and zoning regulations. A growing corpus of interdisciplinary empirical work deals with the diverse aspects of groundwater governance, focusing on formal state-oriented and community-based management practices to knowledge frameworks and socio-environmental justice concerns [20–23].

Historically, high numbers of city settlements were founded along coastal spaces because of their multiple interfacing roles as port cities, military fortifications, navigational nodes and commercial and technological hubs [24]. As a socio-ecological hazard, anthropogenic LS has primarily been associated with rapid unsustainable urban development characterised by high building densities that then result in sediment compaction and excessive groundwater extraction over time [25–27]. Inevitably, the localised impacts of LS across coastal cities have promoted immense infrastructural expenditure and capital-intensive adaptive measures [28].

In this paper we consider the phenomena and effects of coastal subsidence as entangled within multiple ecological, socio-cultural and political concerns and meanings. We also critically consider the multidimensionality of LS, replete with its natural and anthropogenic causes [29]. For example, a singular factor such as groundwater may be a key component but not the only factor affecting LS. Therefore, this interdisciplinary study sets out to explore the following questions: why has political action been slow–or less forthcoming–in redressing some of the alarming rates of subsidence witnessed in Southeast Asia, particularly when considering that densely populated coastal megacities and metropolitan hubs have long been sites of policy intervention and technological innovation? In addition, against more recent policy action, how do municipally-driven mitigation and adaptation practices level up in addressing local realities? How has the governance of LS, as a socio-ecologically multidimensional and multi-sectoral policy conundrum evolved in urban Southeast Asia, if at all? Thus, in exploring these localised complexities in LS, this paper also serves as a rallying call in encouraging more varied work on what we broadly term as *interdisciplinary land subsidence studies*.

## 2. Materials and methods

This analysis combines insights from coastal geography, geodesy, geology, and urban planning. The qualitative and quantitative mixed methods approach integrates interferometric synthetic-aperture radar (InSAR) maps, sea level data, semi-structured qualitative interview data from institutional stakeholders and local residents, together with a content analysis derived from a mapping exercise of core state policies.

### 2.1. InSAR and sea level data

Since the early 1990s, satellite aperture radar (SAR) technology has allowed for interferometric SAR (InSAR) mapping of vertical (and horizontal) surface changes of larger areas, providing new insights into spatial-temporal subsidence. In the current study, InSAR maps from satellite images were integrated using Sentinel-1A (launch 04/2014) data of the European COPERNI-CUS programme to derive spatially resolved subsidence signals. The data is freely available, and each point is mapped every 12 days. The average displacement rates and time series were estimated using persistent scatterer InSAR analysis [30, 31]. In Jakarta and Metro Manila, SAR data in descending satellite orbits are used, while in Singapore, data in ascending orbit is used. In Jakarta and Metro Manila, permanent Global Navigation Satellite System (GNSS) measurements were used to constrain the InSAR vertical displacements. In Singapore, however, GNSS measurements were not available, so the average displacement rate of all InSAR points is fixed to zero. To evaluate the precision of the InSAR results, the root mean square (RMS) of the displacement rates were calculated for areas where no displacement would be expected. The tide gauge data are available at the Permanent Service for Mean Sea Level (PSMSL) [32, 33], the University of Hawaii Sea Level Center [34], from the Indonesian Mapping Agency (BIG) and from GFZ [35]. All data were visually inspected for outliers, drifts and jumps. Radar altimetry was analysed using the ADS system of GFZ [36], applying the most up-to-date environmental and geophysical corrections. Altimetric sea level time series have been derived from a combination of different radar altimeter missions starting from 1993.

### 2.2. Fieldwork interviews

The analysed interview data were derived from 34 qualitative semi-structured interviews and 16 in-depth oral history dialogues at the community level during three distinct fieldwork phases spanning February to June 2017, April and May 2018, and March 2020. Of these, 24 interviews were with representatives from state authorities in key sectors, including urban and

coastal planning, water management, meteorological services, scientific observatories, land reclamation, port development, national housing, Disaster Risk Reduction (DRR) and transportation and public infrastructure. Additionally, discussions were led with a trilateral crossministerial maritime commission, local environmental and pro-poor non-governmental organisational (NGO) networks, senior local scientists (oceanographers, hydrologists, coastal engineers, urban planners and urban sociologists) and three international scientific advisory consortiums in Singapore and Jakarta.

The community-based FGDs were conducted in local dialects at the village level, comprising of *barangay* (Metro Manila) in June 2017 and *kampung* residents and local leaders (Jakarta) in February 2017, and again in March 2020. The small-scale urban units were chosen because of their status as hotspots of risk that received as much state intervention as media attention. The sites spanned four residential spaces in Metro Manila and three communities in Jakarta. Informed verbal consent was sought at all times. Where permission was granted, interviews were voice-taped. All participants at both state agency and community level were anonymized, while guaranteeing confidentiality. The field data were supplemented with a policy mapping for all three cities, focusing on relevant sectors, including urban water supply and freshwater management, urban land use and housing, building regulatory frameworks and related zoning practices.

## 3. Conceptualizing the multidimensionality of land subsidence

The intractability of subsidence as a socio-ecological problem has often been blamed on policy inaction, or what authorities 'choose to do or not to do' [37]. In contexts of policy inaction, emphasis is often placed on empirically tracing and studying the interplay of non-events, nondecisions, denials of agenda and process [38]. These aspects combine with questions of policy reticence [39], indicating instances of risk and blame aversion, cognitive blind spots, institutional paralysis (rooted in sheer inability) and differing perceptions of 'necessary', exigent, or minor socio-ecological problems [40: p. 169].

Narrative approaches have shown how storylines, causal constructions, and emergent realities co-shape how issues are evinced [41], while nascent scholarship in the political sciences have begun to consider questions of affect, particularly in terms of how low and high emotional valence determine policy in/action [42]. While a critical engagement with diverse policy approaches to studying inaction goes beyond the scope of this paper, we argue that articulations of 'inaction' prove useful as a starting point with which to analytically unpack localised LS as a purported wicked problem that policymakers often refer to, and what socio-ecological and political implications this labelling offers.

### 3.1. Land subsidence as a 'wicked' policy problem

As an analytical notion, the 'wicked problem' has emerged as a concept used by diverse actor groups, while facilitating communication across disciplinary and science-policy divides. Coined by design theorists Horst Rittel and Melvin Webber [43], the term 'wicked problem' was used to describe complex, chronic and intractable social issues that pose planning challenges because of their lack of clarity (e.g., issue definitions and causalities), and the fact that they could not be resolved by linear conventional approaches owing to their embeddedness in complex systems and characterized by high degrees of conflicting goals and vested interests among a multiplicity of stakeholders. Furthermore, their purported solutions are likely 'incomplete' or may spur further sub-problems and intended consequences.

In untangling the 'wickedness' of problems, McConnell [40] identifies several crosscutting characteristics that define their intractability. Wicked problems are often perceived as complex

and chronic because every problem is unique and may require diverse approaches to solve them. Furthermore, wicked problems offer diverging value judgements on causalities and the assigning of responsibility. Finally, if wicked problems call for an infinite range of solutions, no point may be reached at which one may claim that a range of options have been exhausted [40].

Levin et al. [44] advanced the concept to draw attention to what they termed as "super-wicked" problems (e.g., the climate crises) entailing four additional dimensions, the fact that: a) time is running out, b) those causing the problem are often unable to seek a solution within their own limits and narrow interests c) the non-existence of a central authority or a weak institution that addresses the problem, and finally d) irrational discounting that pushes the burden of solvability into the future. At first glance, the wickedness or super-wickedness of these socio-environmental dynamics is derived from the fact that in order to arrest subsidence rates, concerted policy action is required. Yet against locally measured rates of land subsidence that are highly variable, the analysis considers a host of reactive policies and adaptive practices that have been put in place, prompting us to revisit the question of whether LS bears the characteristics of a quintessential wicked problem in the first place, and with what implications.

## 3.2. The ambiguities of coastal land subsidence

A key characteristic that has contributed to the ambiguity of politically acting against LS is that it is both a naturally and anthropogenically evolving process. Its natural causes include tectonics, glacial isostatic adjustment and natural sediment compaction [45]. Taking, for example, Venice, the causes of its severe rates of subsidence have been both natural (ground–sea movements) and anthropogenic, with excess groundwater withdrawal amounting to an elevation loss of approximately 23 cm [46]. Geoecological processes such as the stability of marsh land and marsh formation, groundwater flow, vertical and horizontal land deformation or peatland oxidation further add to its multifaceted nature, and thus its causal ambiguity [47–50].

Apart from direct groundwater extraction [51], a number of extractive activities involving oil, gas and coal mining have historically contributed to high subsidence rates [52]. However, what makes coastal cities particularly susceptible to LS, especially those located in delta plains, is that high building densities cause additional compression in shallow layers of 1–20 metres. A physical characteristic of deltaic coasts is that they are comprised of alluvial sediments (alternating between clay, sand and peat) that are subject to 'natural' compaction processes, making urbanised and densely populated spaces more risk-prone.

Furthermore, large-scale construction activities require site dewatering for excavating the buildings' foundations, which not only lowers the water levels, but also results in further land compression. Negative feedback loops further exacerbate the incidence of LS because the sediment delivered along rivers can no longer replenish coastal spaces, instead becoming trapped upstream through damming activity after being dredged or extracted as resources for construction [45].

## 4. Results and discussion

This first section reviews how methodologies for measuring and monitoring coastal changes have historically evolved. It then considers current figures and projections of sea level change in the three cities, before linking these with land subsidence rates. Based on these reflections, we argue in favour of analysing LS as a politically ambiguous notion and a differentially experienced phenomenon in everyday life. The analysis proceeds to assess contemporary mitigation and adaptation practices to urban coastal subsidence, together with local community perceptions.

## 4.1. Figures, projections, and historic path dependencies

Threats to coastal areas related to sea level change can be evaluated by looking at two main factors: changes in the regional sea level trend and coastal subsidence. Sea level rise (SLR) is largely driven by climate-related changes and currently has a global average rate of 3.2 mm per year [e.g., 53]. In Southeast Asia, SLR is superimposed by annual sea level variations of up to 40 cm, which are sometimes governed by events like El Niño and La Niña. Sea level has been measured and monitored continuously and globally by space-based radar altimetry since 1992. Along coastlines, sea levels are traditionally measured by tide gauges, which provide tidal ranges, surges and changes of relative sea level at a specific point. Some tide gauges have a centennial history (e.g., Manila starting in 1901), delivering invaluable information about past trends. Since 2000, an increasing number of tide gauges have been equipped with continuous GNSS for vertical height control [54], making this data comparable to radar altimetry (e.g., Jakarta).

The second factor is LS itself–a phenomena that is both natural and anthropogenic. In the past, subsidence has been measured using classical repeated levelling along selected lines and, since the early 1990s, with GNSS. Both technologies allow for the estimation of vertical displacements at discrete points but miss some of the spatial patterns. Combining the spatial InSAR information and the point-wise GNSS provides subsidence information that is consistent among different geodetic observations.

**4.1.1. Metro Manila, Philippines.** Metro Manila offers to be taken as a prime example of human-induced subsidence. Between 1901 and 1965, the tide gauge record has shown a trend of +1.5 mm/a, increasing to +14.7 mm/a afterward. The earlier rate can be associated with the long-term global SLR. The cumulative water withdrawal [55] after the 1940s increased the effect on local SLR only slightly, while the rapid change after 1965 is likely to be associated with increased ground water withdrawal. Between 1961 and 1967, the Angat Dam was built along with some other hydropower constructions downstream from Metro Manila, changing the water supply scheme. A significant increase in groundwater withdrawal occurred between 1970 and 1980, going up about four times the volume extracted in 1970 [Fig 2 in 55], explaining the change in the tide gauge trend. For the period of 1970 to 1994, the increase in tidal level (to be interpreted as subsidence) seems to closely follow the changes in water withdrawal. From 1980 to 2009 the amount of extracted water remained on a (stable) high, causing a stable tide gauge trend. After 2009, tide gauge data show a deceleration with a lower trend of 5 mm/a.

The regional sea level west of Metro Manila (South China Sea, SCS) was also measured using radar altimetry; which is characterised by a net SLR of 4.0 mm/a and an annual amplitude of 40 cm. Additionally, the altimetric sea level record reveals decadal variability and the influence of El Niño and La Niña patterns. Because of its bay characteristic, the sea level trends and seasonal signals might be slightly different. Summarising the general picture of the sea level records are in good agreement: prior to 1965, the tide gauge sea level trend could be explained by SLR, while the post-1965 trends need to include coastal subsidence.

For the period of 2014 to 2017, subsidence in Metro Manila was estimated with InSAR using Sentinel-1A data (Fig 1). Most rates are negative (subsidence) and range from a slight uplift (4 mm/a) for Capital District and south and east of Capital District to as much as 170 mm/a subsidence in the north of Metro Manila (Guiguinto), see Table 1. Also west of Laguna de Bay, a few areas can be identified (Dasmariñas, Canlalay, Casile, Santa Rosa City) with higher subsidence rates (approaching 100 mm/a). The subsidence rates at Balagtas (130 mm/a), Guiguinto, Malolos City (100 mm/a) or Calumpit (140 mm/a) agree well with the urbanisation seen there in recent years [56]. The comparison with earlier studies of LS in Metro Manila

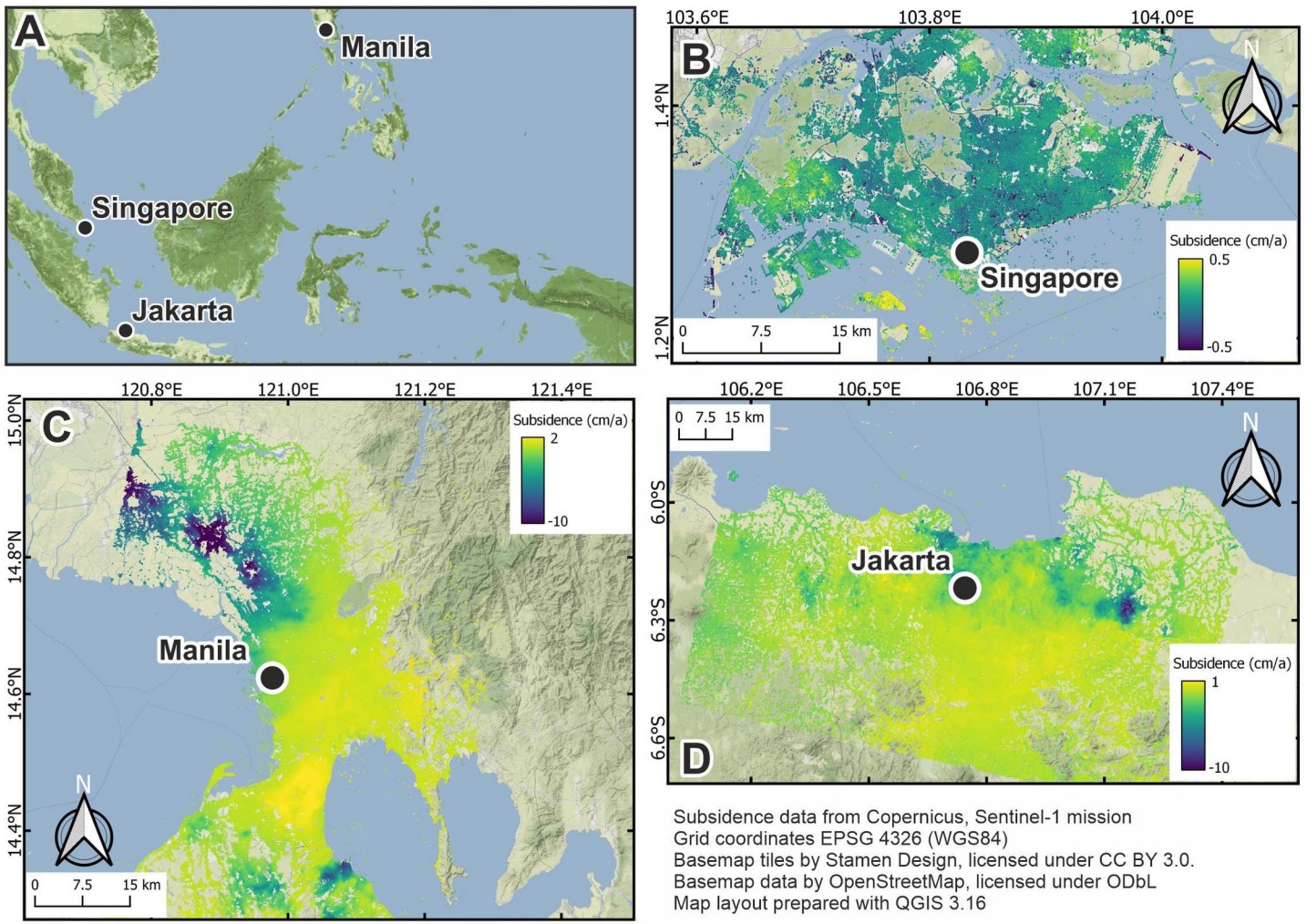

**Fig 1.** Vertical displacement rates for Jakarta (D), Metro Manila (C) and Singapore (D). Note that Singapore's displacement is scaled to -2/2 cm/a. Contains Copernicus data 2018.

[e.g., 57; InSAR data available at http://insarmaps.miami.edu] shows a high spatial and temporal variability of the subsidence pattern. In the northwestern part (Valenzuela, Meycauayan, and Malolos), the rates are highly variable over time. These changes may be attributed to the intensification of urban development [e.g., 56]. The current subsidence rate for the area around the tide gauge derived from InSAR (average of the closest four points) is ~6 mm/a (10/2014–10/2017), which is lower than the 10.7 mm/a (LOS) reported by Raucoules et al. [57].

 **4.1.2. Jakarta, Indonesia.** LS in Jakarta was observed for the first time in 1926 by Dutch surveyors who repeated the levelling lines in metropolitan Jakarta [58]. In 1978, new evidence

**Table 1. Main characteristics of SAR images used in InSAR analysis.**

| Study area | Orbit | Time span | Number of images | Incidence angle |
|---|---|---|---|---|
| Jakarta | Descending | 20.11.2015–20.01.2018 | 43 | 37° |
| Manila | Descending | 19.10.2014–15.10.2017 | 79 | 39° |
| Singapore | Ascending | 29.10.2014–11.03.2017 | 66 | 43° |

of subsidence was observed by cracks in a bridge. Following this incident, a network of levelling points was established and re-measured in 1982, 1991, 1997 and 1999 [59]. Later, a network of GNSS points was established and measured through periodic campaigns and, recently, some sites in a continuous manner [60]. The regional sea level near Jakarta (over a 25-year period) has shown a positive trend of ~4.3 mm/a, slightly above the global mean. Prominent events causing dynamic sea level changes such as El Niño or La Niña [e.g., 1997 and 2011] are clearly visible in the record, but their effects do not alter the general positive trend. On multiyear time scales, the sea level trend is slightly influenced by decadal variations. The annual signal is small, with amplitudes of less than 20 cm. In the city of Jakarta, three tide gauges are operated—two in the subdistrict Tanjung Priok and one in the Penjaringan subdistrict. Because of near-coastal processes and the harbour location of the tide gauges, the sea level rates derived from these sensors do not always give identical results to those derived from radar altimetry.

The subsidence rates along the Jakarta coast have been found to be highly variable. Averaged over 10 years, the area of Penaringan shows rates as much as 85 mm/a, and in the Tanjung Priok area, around 40 mm/a [61]. In our study using Sentinel-1A (Fig 1), the maximum subsidence rates along the coast are currently less than 60 mm/a, with most areas showing rates of less than 25 mm/a. A few areas show higher subsidence rates, in particular the area around Dadap Kosambi with ~70 mm/a, Duri Kosambi (Cengkareng) with 60 mm/a and Sagara Makmur area with ~65 mm/a. Kota Bekasi, with rates of ~40 mm/a, seems to be developing as a new subsidence hotspot area. Cikarang Utara, outside of Jakarta, has the highest subsidence rates with more than 100 mm/a. The study of Chaussard et al. [62] using ALOS InSAR shows rates of more than 200 mm/a in some local areas but with a high degree of spatial variability. Areas in this previous study showing high subsidence (e.g., Muara Baru, Cengkareng) are now subsiding at a lower rate of about 60 mm/a, which can be traced to the relocation of factories out of this area. In summary, comparing our study with previous analyses [59, 62–64], subsidence levels in Jakarta and its surrounding area show high spatial and temporal variability.

**4.1.3. Singapore.** Far surpassing the monitoring resources of Metro Manila and Jakarta, Singapore possesses 13 tide gauges placed along its national coastal borders. The earliest available records started in 1954. A few tide gauge sites have undergone major hardware upgrades, preventing the concatenation of consecutive data into a single continuous time series. The majority of tide gauges (nine) show positive sea level trends between 1 and 6 mm/a, while four tide gauges show a small drop of about 1 mm/a. Comparing these tide gauge trends with the nearby InSAR vertical rates does not give evidence of larger discrepancies in the observed values.

Singapore's local sea level is dominated by two different ocean regimes–to the west by the Malacca Straits and to the east by the SCS, both of which are connected by the Singapore Straits. The SCS altimetric sea level record displays a strong annual signal with amplitudes of 40 cm. The trend is +2.5 mm/a superseded by a variation associated with the Southern Oscillation Index (SOI). The sea level trend in the Malacca Strait is around +4 mm/a but is dominated by a short period variability (60 days) with amplitudes of 40 cm but missing the annual signal.

Our InSAR studies (Fig 1) of the subsidence patterns show only small changes. In the past few decades, Singapore has undergone large-scale land reclamation, especially extending its area to the south-east (Singapore's Changi International Airport) and to the south-west (harbour areas), consequently experiencing subsidence. Most of the inner-city areas show no subsidence. A few areas of localised subsidence have subsidence rates of 10 mm/a, which can be traced to construction activities. The exceptions to these small rates are the aforementioned reclamation areas, where the subsidence rate is as much as 15 mm/a. Catalão et al. [65]

performed an InSAR analysis for the period 1995–2000. The pattern is similar to our more recent estimates, leading to the finding of a generally small and homogeneous spatial and temporal subsidence pattern. This phenomenon could be explained by the geological nature of much of the island (particularly towards the west and its centre) is composed of sedimentary and igneous rock [66]. Smaller parts of the city, for example, the Kallang formation on which high-end real estate development has progressively been taking place, rest on younger, less-compacted sediments with the potential for subsidence.

Taken together, our SAR datasets illustrate subsidence rates with high degrees of local variation.

## 4.2. Urban subsidence histories in perspective

In a contemporary context, Jakarta remains an internationally politicized and an oft-cited example of a climate-induced "sinking city". Coastal retreat and urban resettlement are prominently featured in its public discourses, particularly against a backdrop devastating floods, the last of which occurred towards the end of 2019. The Central Government announcing the relocation of Jakarta's administrative organs over a span of three decades bears testimony to the socio-ecological identity of its fast subsiding northern coastline, often referred to as its "last frontier". Unlike in Jakarta, where the politicised narrative of the sinking city remains all too visible, discourses on disasters and hazards in Metro Manila remain anchored in its complex volcanic, earthquake- and flood-related politics, in which policymakers often self-define Manila as a quintessential "Disaster Capital".

Both Jakarta and Metro Manila refer to their precarious position within the "Ring of Fire", as insular Southeast Asia is regularly depicted by the presence of volcanic activity and tectonic shifts. In Singapore however, with the exception of warming urban temperatures and heat islands, sea level change has up until recently remained a non-issue. However thematic references made to the affluent island-state's environmental precarity during the Prime Minister's Address during its National Day celebrations in 2019, offers to be taken as a significant shift in policy discourse. Singapore's solutions to creeping sea level change and coastal erosion has largely been technological in nature, finding its place as a fast-paced regional economic and knowledge hub that is both materially and metaphorically secure and "fortified".

Furthermore, the three Southeast Asian cities of Jakarta, Metro Manila, and Singapore share distinct colonial trajectories and coastal identities. The Philippines was a former Spanish and latterly American colony; Jakarta was hailed as the 'jewel of Dutch Batavia', and Singapore was formerly a key port city of British Malaya, following its subsequent independence from Malaysia in 1965. Hydraulic management remained a distinct feature of colonial intervention in all three cities, in which the Dutch built a canal system through Jakarta and the Spanish in Manila with its *esteros* riverine network [67]. Furthermore, Singapore's first reclamation activity began in the late nineteenth century, starting with the expansion of Telok Ayer followed by Collyer Quay (subsidence rates of less than 2 mm/a) and the widening of Singapore River to enable easier port access [68].

The extraction of groundwater has been the key cause of LS in both Metro Manila and Jakarta. The massive groundwater usage to fill aquaculture ponds were cited during fieldwork as early causes of subsidence in coastal Jakarta and Metro Manila. This issue was further compounded by state intervention in supplying groundwater pumps to meet deficits in the supply of pipe-borne water, thereby creating dependence on subsurface water resources as a temporary fix to inadequate urban water provision [69]. Industrialisation and the expansion of low-income informal housing settlements were severely underserviced regarding sanitation and waste disposal. These new urban spaces were perceived to put further pressure on the quality

of available freshwater sources because of surface pollution by industrial and agricultural effluents.

The municipal water supply in both cities has been hindered by several demand-side aspects. The first entailed perceptions regarding higher tariff rates and other consumer costs. In Jakarta, for example, the combined effect of the privatisation of water supply, leading to additional connection fees (amounting to a month's minimum wage on average), and rising bills made pipe-borne water more expensive than maintaining a backyard well [70, 71]. Second, our field observations in northern Jakarta revealed that fewer residents trusted the quality of tap water because much of it was sourced from several polluted water bodies, including Jakarta's Citarum River. Approximately 90% of all residents depend on decentralised sanitation that lacks proper wastewater treatment, groundwater is increasingly found to be contaminated with fecalcoliform bacteria [72 p. 899].

Past estimations for the Philippines suggest that around 86% of the piped water supply is groundwater, 63% of this is consumed by the domestic sector [73]. Furthermore, it was estimated that 44% of Metro Manila's inhabitants have access to direct house connections, and those without access depend on wells, communal faucets, springs and small-scale providers [74]. In addition, the Philippines experienced a series of 'water wars' [75], illustrating the criticality of its surface water and groundwater resources in supporting rapidly urbanising spaces–with potentially aggravating consequences for local subsidence rates. These struggles have continued against the backdrop of large-scale corporatisation and privatisation initiatives, spurring conflicts with operators such as Nestlé Philippines in Laguna [73]. Before the subsequent privatisation of the Metropolitan Waterworks and Sewerage System (MWSS) in 1997, disruptions in water service and quality took place. The two private operators that were ultimately awarded 25-year concession agreements–Maynilad Water Services Inc. and the Manila Water Company–were also tasked with undertaking costly rehabilitation work. An increase in water supply tariffs by the two concessionaires subsequently led to a long-drawn international arbitration process [75]. These developments affected local (informal) pumping activities and thus land subsidence rates until today.

Taken together, both Jakarta and Metro Manila appear to be similar when considering the implementation of large-scale infrastructural anti-flood projects. Examples include the coastal KAMANAVA area in Metro Manila, and the building of sea walls in Jakarta, with injections of development loans and technology from Japanese and Dutch consultancies [76, 77]. The nature of these capital-intensive interventions have arguably prompted a vicious cycle of rising damages to buildings, transport infrastructure and other public works caused by LS. Moreover, frequent flooding occurs when rain drainage to the sea becomes increasingly difficult due to the raising of dykes and flood walls, together with bridges and other subsurface structures.

When compared with Metro Manila and Jakarta, Singapore remains an outlier as LS rates are comparatively insignificant. 70–80% of Singapore's coastline is fortified with walls and stone embankments to curtail tidal erosion [78]. Although the entire city is served by a stable pipe-borne water supply regulated by the Public Utilities Board (PUB), Singapore's 'four tap policy' providing 1.6 million cubic meters of water per day [79 p. 7], has been hailed as a success story. However, the LS rates revealed by our InSAR maps show temporal variations that span historic reclamation activity, largely in the southern and western fringes of Singapore. The scarcity of natural resources, particularly land and water, has been a primary leitmotif in Singapore's nation-building narrative [80]. Two primary forms of large-scale urban development activity characterise its urban form: coastal land reclamation from the sea and underground tunnelling, particularly for its extensive subway mass rail transit system, both of which have been associated with very localised forms of subsidence.

## 5. Mitigation and adaptation to urban coastal subsidence: Contradictions in public policy and community practices

In investigating the scale and implications of municipal and community-level responses to LS, a preliminary question that needs to be explored is *how and why* land subsidence first entered public discourse, particularly regarding policy nomenclature on urban flooding [81]. Sunken spaces may constantly be inundated with polluted and vector-borne stagnant water. Notwith-standing, in Jakarta, as well as in Metro Manila, scientific discourses on subsidence have squarely pointed to groundwater exploitation as a primary cause [82, 83]. In Singapore, LS hardly appears as a subject in everyday environmental politics and media reports, also because Singapore positions itself as offering one of the region's most technologically sophisticated yet centrally governed models in the management of freshwater resources. Thus in Singapore, LS remains a non-issue, even in closely integrated and centrally regulated construction codes, groundwater mining policies, zoning practices and land reclamation specifications.

### 5.1. The multi-sectoral nature of mitigation policies

**5.1.1. Groundwater and surface water resources.** No integrated national groundwater protection policy exists in Metro Manila and Jakarta. In the Philippines, groundwater falls under the aegis of the Philippines Water Code that stipulates 'first in time priority in right' and bundles the ownership and use of groundwater to the land above it. In particular, the Philippine Clean Water Act 2004 (under the Department of Environment and Natural Resources/ DENR and the National Water Resources Board/NWRB) governs both surface and coastal water use, whereas groundwater falls under the Philippines National Standards for Drinking Water due to its foremost concern with *salinity* rather than pollution [73]. In effect, there are 30 state departments and agencies dealing with the disparate aspects of urban water management in Metro Manila, from flood control and irrigation to hydropower and watershed management [84].

Early policies in Metro Manila indirectly allowed for the tapping of aquifers through wells given the uneven access to urban water as a secondary source. Subsequently, an amendment to the Water Code (under Presidential Decree 1067) prohibited the construction of deep wells within municipal limits, but groundwater continued to be depleted as a result of little monitoring and enforcement capacity at the city council level. The enforcement of sanitation laws within the Water Code was devolved to local government units, regulatory action was arguably, at best, a patchwork of fragmented remedial measures across Metro Manila's 16 cities and one municipality. For example, Quezon City's (no significant subsidence) council passed Ordinance 1682-S-2005 that prohibited the drilling of new deep wells while enabling the regulation of existing ones, with penalties imposed on violators, such as the sealing of illegal wells [84]. In Metro Manila, the NWRB placed as much spatial emphasis on commercial hubs such as Makati City (no significant subsidence) as on its large business establishments and highrises [85, 86].

As municipal freshwater supply was deemed costly, dependence on groundwater sources increased with over 58% of the water demand being supplied from sources deemed either 'unaccounted-for-water' or 'non-revenue water' from the very beginning [87]. In addition, the state-led action of reducing water waste together with curtailing industrial pollution proved non-effective [73]. Thus, uneven municipal freshwater access, further compounded by the salinization and pollution of freshwater and groundwater resources, with the extraction of the latter being further criminalised, led to a vicious cycle of non-legal extraction and underreporting.

The evolution of Jakarta's groundwater governance bears a similar trajectory to that of Metro Manila. Although subsidence-related issues are now entrenched in Jakarta's everyday public discourse, the efficacy of mitigation efforts have not been proven in the eyes of its public. Groundwater extraction was identified as the main cause of LS in Jakarta as early as 1991. At the time, Governor Wiyogo Atmodarminto made the construction of infiltration wells mandatory in order to obtain building permits (latterly revised as Regulation No. 20/2013). Furthermore, disincentives for groundwater extraction were put in place through taxation, particularly targeting subsurface construction. Yet, the effectiveness of such policies rested on their enforcement capacities, political will, and readiness in supporting alternative forms of clean freshwater provision. Practices such as rainwater harvesting emerged as a feasible solution, whereas technology such as desalination remained far too costly to adopt. In 2009, the Ministry of the Environment began focusing on restoring the groundwater table, issuing a decree (Regulation No.12/2009) requiring newly constructed commercial buildings to store rainwater in retention ponds, infiltration wells and/or deep biopore cylinders.

Given the adverse impact LS has on urban flooding, subsidence mitigation efforts were integrated into flood control programmes [88]. The Jakarta Coastal Defence Strategy (JCDS) emerged as the first state-funded attempt at integrating subsidence studies in the metropolis [45]. Thus, the western and eastern canal system and several retention ponds were developed through its Flood Control Infrastructure Project. In 2012 the JCDS was changed into the National Capital Integrated Coastal Development Master Plan (NCICD) led by the Central Government, and included a Giant Sea Wall (GSW), a sub-project designed to address land subsidence with a 65-year timeframe.

However, local scientists argue that current mitigation policies are not supported by a system that continuously monitors LS rates. It is often argued that monitoring instrumentation such as extensometers and piezometers should be provided in badly subsided areas, as well as newly reclaimed coastal spaces [89, 90]. The creation of a special intergovernmental taskforce to oversee the management of LS in a multi-sectoral manner has often been raised in Jakarta's public policy meetings.

In Singapore, the discourses related to groundwater have taken a different turn. With the centralised provision of pipe-borne water within the island managed by its PUB, its Four National Taps Policy combines water from local catchment areas, imported freshwater, desalinated water and recycled water referred as NEWater [91]. Water has remained a core aspect of national security, particularly regarding Singapore's postcolonial dependence on freshwater sources from neighbouring Malaysia. More recently, discussions have been held about creating a 'fifth' tap, in the form of a geo-engineered aquifer that would make use of natural underwater cavities to store rainwater. The exploration is led by PUB with a focus on the Jurong Formation in the western part of the island, largely comprising porous rock and sandstone [89]. In public media, the 'fifth tap' is positioned as a 'sustainable water source . . . by creating a cycle of pumping water out at one end, and allowing rainfall to replace it while giving time for the groundwater to be naturally cleaned before being pumped out again' [92].

**5.1.2. Building codes and zoning regulation.** Apart from the overexploitation of groundwater aquifers, the compounding effects of sediment compaction on LS have barely been explored in policy-focused scholarship. Anthropogenic compaction can be caused by the loading of built infrastructures, particularly vertical buildings on spaces that are already subsiding. Compaction inevitably leads to further infrastructural damage, roads, railways, bridges and subsurface networks such as sewerage and water pipelines. Building regulations do not always take these connections into account. In Singapore, the integrated nature of its Building and Construction Authority's (BCA) Building Control Act of 2007 places much emphasis on subsurface constructions during tunnelling activity for example [89, 93]. Yet the Act does not

explicitly mention subsidence as a consideration in underground building design. In Metro Manila, infrastructural development falls within the aegis of the National Building Code of the Philippines (Republic Act No. 6541), which identifies low coastal elevation zones (LECZs) as 'danger zones' by its Disaster Risk Reduction units.

However, subsidence sensitivity does not feature in its existing building codes and regulations that are often–in the perspectives of local policy makers in the Philippines–arbitrarily adhered to and enforced. Furthermore, integrated coastal land use zoning practices often restrict the number of high-rises, while regulating commercial and industrial activities. In Singapore, land zoning for the construction of high-rises falls under the mandate of its Urban Redevelopment Authority through its Master Plan of 2014. Metro Manila implemented its Comprehensive Land Use Plan and Zoning Ordinance of 2006/Ordinance 8119, which integrated water-use policies with a greater emphasis placed on water quality and shallow groundwater resources [94].

In Jakarta, zoning regulations are used as a preventive tool in restricting infrastructural loading and groundwater exploitation. The Detailed Spatial Plan (Perda DKI No. 1/2014) ushered decentralised zoning that has enabled progressive forms of experimentation. For example, a moratorium on the construction of new malls was put into place; however, the regulation failed to stipulate building heights [71]. Yet some may argue that northern Jakarta should reduce the construction of massive buildings and force the industries to relocate to peri-urban areas. In recent years, the relocation of particular factories and warehouses in Jakarta induced a decline in localised subsidence rates, however these efforts have been piecemeal. The Detailed Spatial Planning and Zoning Regulation is expected to be an initial instrument to establish spatial and time limits to industrial uses and increase the proportion of green and blue open space [see 95] in the land subsiding area. In recent years, planners in Jakarta have been considering the translation of the Sponge City concept as utilised in spaces across China [e.g. Lingang District in Shanghai, Wuhan], Philadelphia, and Taipei [96]. With the hope of capturing lost surface-run off and reducing urban flooding [97], these nascent plans call for the construction of bio-retention infrastructure, surface-water infiltration trenches, and expanding on artificial wetlands. Yet, solid waste pollution and industrial effluents that compromise loaded drainage systems remain an embattled question.

**5.1.3. Evictions of informal settlements and the politics of public–private partnerships.** Spaces that are characteristic of both marginality and affluence constitute a typical feature of coastal megacities in the Global South. Furthermore, the privatisation of shored areas with real estate potential because of their waterfront aesthetics has remained an enduring feature of contemporary urban transformation. At the same time, media narratives deploying terms such as 'runaway development' in the context of subsidence [17] reveal great ambivalence regarding the contradictory processes of coastal densification; this densification can be seen as a result of unplanned urban sprawl, in which 'poverty pockets' in low-elevation flood-prone areas are paralleled with high-end property development.

The emphasis placed on clearing 'squatter' settlements has remained a tactical mainstay in urban politics within Metro Manila and Jakarta alike, resulting in centrally planned processes of evictions. Poverty pockets have often been blamed for urban flooding because of their encroachment on freshwater channels and marshy coastlines, an issue compounded by the visibility of solid waste blockages in these areas [98, 99]. Moreover, state relocation projects have been notorious for giving way to public–private partnerships (PPPs), often under city-level patronage, in areas deemed as vulnerability 'hotspots' because of recurrent flooding that have been cleared and subsequently primed for lucrative real estate development [100].

**5.1.4. Disaster risk reduction practices and monitoring.** One of the most easily overlooked policy aspects of states' response to subsidence has been the relationship between DRR

processes and the monitoring structures and methodologies that have been put in place for policy implementation. It is thus worth noting that while the municipal provision of water pumps across Metro Manila and Jakarta worsened groundwater extraction, the establishment of several coordinating institutions (e.g. National Disaster Risk Reduction and Management Council in Metro Manila) and the issuing of national action plans (e.g. Strategic National Action Plan and the National Disaster Management Plan 2010–2014 of Jakarta) indicate an increased awareness of the challenging situation. Singapore's artificial Semakau Island for instance hosts one of the nine stations of the Satellite Positioning Reference Network (SiReNT) The stations carry out subsidence measurements and the monitoring of tectonic movements, together with other real-time applications such as surveying, cadastre and car navigation (interview with a geologist at the Tropical Marine Science Institute, June 2017). Hazard-related events such as storm surges, earthquakes, super-typhoons are being anticipated and the need for policy responses acknowledged.

## 5.2. Adaptive practices at the municipal and communal level

**5.2.1. Existing infrastructural fixes and urban coastal innovation.** In both Metro Manila and Jakarta, the most visible form of urban adaptation can be seen in the numerous costly coastal protection and flood control structures, often critiqued as being ineffective. Jakarta raised its 30 kilometers of seawall, and the structure continues to be raised above the height of waves as subsidence continues to lower the city's foundational levels [88]. Northern Manila´s extensive multi-staged KAMANAVA flood control programme includes the recent construction of a polder dyke, fraught with public concern regarding the appropriateness of its height and seaward access points [101]. These infrastructure-heavy 'palliative' measures [25] further glosses over long-term systemic solutions such as urban water retention potential, including rainwater harvesting.

Yet for Singapore as well, raising the level of coastal infrastructures remains the most commonplace state-endorsed practice, as has been the case since 2011 for both minimum platform levels and minimum levels of reclaimed land through the Code of Practice on Surface Water Drainage [91]. Singapore's Climate Action Plan [2016] issued by its National Climate Change Secretariat (NCCS) called for early preparations to safeguard the city-state through expansive engineering work that has an emphasis on coastal defence and stabilisation structures, together with the raising of selected transport infrastructure and its international airport [102]. It is worth noting that Singapore, as a technological experimental hub, has been instrumental in shaping international scientific discussions on urban water resource management, including the development of amphibious architecture and large floating structures [103], particularly in countering its costly and politically contentious land reclamation activity [104].

**5.2.2. Foreshore and island reclamation.** Land reclamation as a practice has a long history in archipelagic Southeast Asia, starting from early colonial interventions in both Singapore and, to a lesser extent, Dutch Batavia. Singapore's land extent expanded by approximately 25% over the past two centuries [68]. Reclamation work continued along the Bay of Jakarta and Manila Bay under the regimes of both Soeharto and Marcos, with considerable waterfront expansion in spaces such as Parañaque and Cavite in southern Manila. Over the past decade, the return to what Colven [77] refers to as the 'allure of big infrastructure' can be witnessed in the revivalist discourse of foreshore and offshore island reclamation as a primary means of increasing high-value urban land. In Metro Manila, vocal anti-reclamation lobbying was led by key scientists and environmental civil society activists who argued against projects such as Sangley Point's new airport (subsidence ~3 mm/a) on grounds of increasing soil liquefaction

[25], a phenomenon in which the stiffness and strength of ground soil is reduced by movements caused during earthquake activity and other forms of rapid loading.

Among the numerous ambitious land reclamation projects sits northern Jakarta's National Capital Integrated Coastal Development Master Plan (NCICD) that envisions a giant seawall. Original sketches of the plan had the project resemble a great Garuda, Indonesia's mythological national icon, when viewed aerially. The original master plan included the refortification of the existing sea wall and the construction of a 56 km outer seawall enclosing Jakarta Bay, flanked by 17 new artificial islands. Critics have argued that the project would do little to restore the flow of the 13 rivers that feed into the sinking city and to mitigate the current rates of subsidence along its existing coastal fringes [77]. This criticism has not led to a fundamental reconsideration of the project [105, 106]. Thus, plans for fashioning new 'land from sea' could be framed as an adaptive measure in the name of subsidence and relative sea level rise, but would do little to address the root causes of LS. While further reclamation work was ceased in October 2018 by the newly elected Governor of Jakarta under grounds that developers had failed on conducting appropriate environmental impact assessments and for tax evasion, four islands had been completed. Over the course of 2019, developers allegedly continued construction particularly in Islet D, despite having to wait for the passing of a new zonal bylaw [107]. Since Jakarta's COVID-19 crisis, further construction has momentarily ceased. Further attempts at curbing subsidence in Jakarta are now being supported by a newly assembled expert panel funded by the Japan International Cooperation Agency (JICA), which will work on identifying policies that require contextual embedding in a fragmented governance system, in which so-called 'best practices' from cities such as Tokyo, may only be selectively translated.

**5.2.3. Small-scale adaptive practices at the community level.** Community-level household interviews indicated that the primary focus on adapting to storm surges and inland flooding [108], while the phenomenon of LS was barely visible. In the *kampungs* of northern Jakarta, however, community-level adaptation practices were largely infrastructural in nature, and appeared to be mostly self-financed and implemented at the household level. Kampung leaders indicated that modest household contributions would be pooled to finance a new well or to add height to an existing state-built flood wall that had sunk [88]. Furthermore, the *kampungs* in northern Jakarta continue to practice a small-scale form of reclaiming shallow coastal water, locally referred to as *nimbun*, for expanding land out into sea by a few metres to less than half a kilometre. This is often achieved by piling wood debris and other construction material until shallow sand bed surfaces become hardened by tidal activity and sedimentation.

What remains understudied, however, are the dynamics of coastal lands that are communally deemed unusable and are either a) relegated as landfills and solid waste dumpsites that are semi-permanently inundated during seasonal monsoonal flooding, or b) low-lying areas that witness some form of recurrent inundation when roads and public infrastructure in close proximity are raised, for example, as in the case of northern Metro Manila's submerged Artex compound–ironically dubbed by local media as the 'Venice of Malabon' [109]. Prior to 2010, this area saw subsidence rates of 70 mm/a [57], while our more recent data show rates of 30 mm/a. Furthermore, DRR measures in providing simple pumps and pump houses at the community level are often deemed insufficient because they are seen as re-circulating volumes of "water from one sunken neighbourhood to another" during flooding. Taken together, Fig 2 illustrates the multi-dimensionality of LS as a natural geological and an anthropogenic process.

Thus, subsidence as a socio-environmental hazard remains bound to multiple urban metabolic flows, not just of freshwater, groundwater and coastal currents. Contemporary transformations in the built environment together with a number of anti-flood adaptation practices are seen to further exacerbate vulnerabilities and create new risks, particularly in low-lying

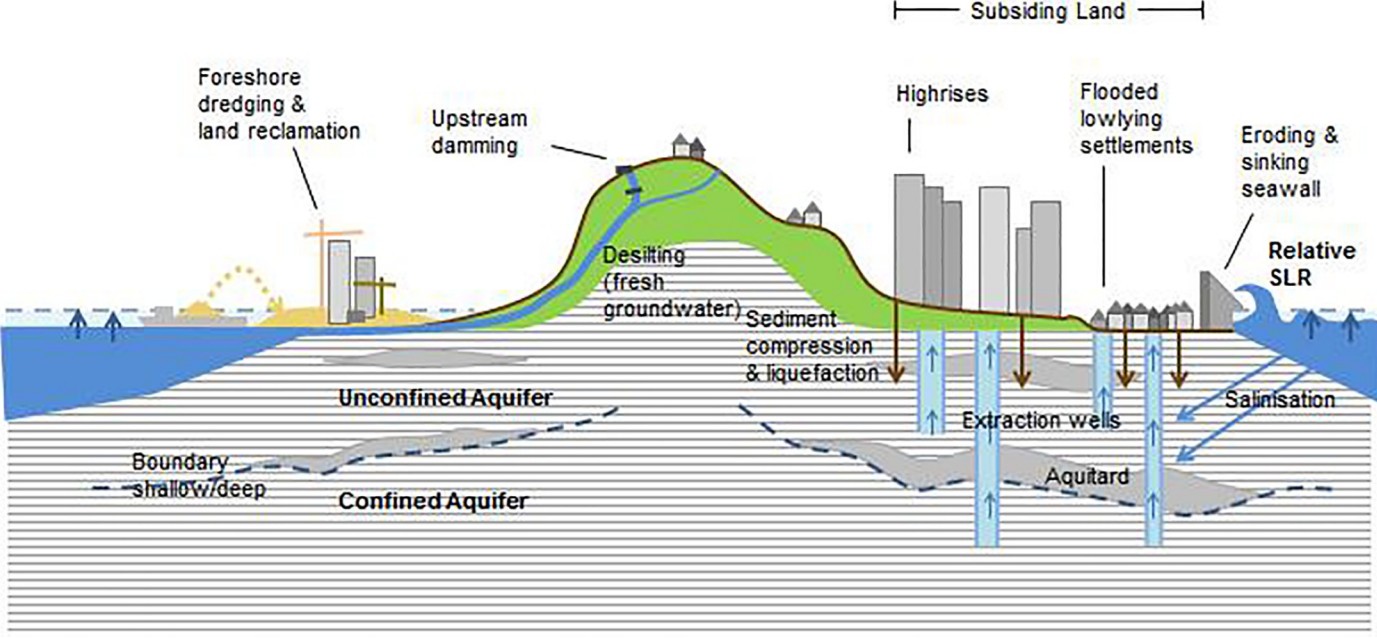

**Fig 2. Conceptual illustration of anthropogenic and natural processes impacting LS in high-density coastal cities (source: Authors own).**

urban littoral spaces doubling affected by the onset of rising regional sea levels. The vicious cycle of infrastructure-led solutions, for example the raising of roads and land reclamation render LS a classic wicked problem. In part its 'wickedness' is owed to its intractability in identifying groundwater extraction as its singular root cause. Second, future interactions of fast subsiding land with other climate-induced coastal hazards such as heavy monsoonal stormwater flooding, slow-onset rising tides and coastal erosion makes it increasingly difficult to argue a case for governing LS as a 'new' phenomenon in itself. As the last subsection illustrates, differing public perceptions and clouded policy framings further play into the 'wickedness' of recasting LS as a governance issue its own right, despite high economic and political stakes in attending to it in reactionary, remedial and/or piecemeal ways.

## 6. Contradictions in public discourse: Reversibility, naturalization, and individualization

As previously established, the two metropolitan capitals of Jakarta and Metro Manila are seen to be among the most at risk of LS in archipelagic Southeast Asia [25, 64, 88]. Similar postcolonial trajectories of rapid urbanisation characterised by industrial agglomeration and the high incidence of rural-to-urban migration resulting in urban sprawl lend to their comparability as case studies. Yet, sea level rise and climate change have been discussed quite differently in both cities, with Jakarta increasingly being recognized as a potentially sinking city with connected adaptation debates, whereas Manila still puts much effort on disaster risk reduction strategies mainly geared towards typhoon events, seasonal flooding and seismic activity. Singapore, a regional model of economic growth and centralised planning, adds to the current study because of its unique position as a metropolitan island state [67, 110], heavily investing into coastal infrastructure projects involving dredging [80, 111], foreshore reclamation and underground tunnelling [112, 113], partly also in expectation of changing sea levels with the aim of installing protective, infrastructure-driven adaptation.

Taking these diverse mitigations and adaptation-led practices together, the three cities continue to embody distinct media-fed identities: Jakarta as the 'world's fastest sinking city', Manila as the quintessential multi-hazard 'disaster capital' and Singapore as regional hub and as a high-tech enclaved and fortified metropolis. While these urban identities differ substantially, contradictions in perceiving LS within public discourse have remained largely similar across the three cities. Despite these differences, three distinct narratives can be found in explicitly land subsidence related stakeholder perceptions: a) the reversibility of subsidence; b) the naturalisation of subsidence; and c) the individualisation of subsidence.

## 6.1. Subsidence as a "reversible" phenomenon in the long term

Groundwater depletion is often cited as a primary cause for LS in all three cities. Although groundwater levels can be measured, the difficulty of communicating about groundwater as a finite resource has often been problematized by policymakers because of its invisibility, making it more difficult to discern the rates of anthropogenic LS. Public discourses on LS often focus on abandoned wells, cracked walls, roadways and sunken compounds, making it all the more salient to analyze the politics of infrastructure through the lens of in/visibility [114]. Despite the severity of the damage to urban infrastructure, the political perplexities of assigning causal factors to anthropogenic subsidence comes from the fact that there are diverse 'styles of knowing' that assign not only credibility and legitimacy [115 p. 149], but also definitional *legibility*, particularly regarding what counts as anthropogenic subsidence. For example in Jakarta, the Indonesian-Javanese term *amblas* refers to sinking land. Household interviews at community *kampung*-level, particularly across coastal settlements in Kamal Muara and Muara Bahru indicated that perceptions on what caused subsidence were diverse and multi-causal. Many attributed subsidence to localized landfill activity, landslides (*longsor* in Javanese), and the raising of roadways. Some cited the fact that decades ago, the more visible indicators of subsidence such as cracked walls were commonly misinterpreted as resulting from inadequate drainage and the "building errors and the lack of concrete" (Kamal Bahru interview, March 2020).

## 6.2. Naturalization: Subsidence as an inevitable hydrological occurrence

As the effects of anthropogenic subsidence gain greater awareness, the notion of LS as a natural phenomenon gains less salience. Yet this belief was said to have mired early political action in Jakarta and Metro Manila. Referred to at times as the 'precipitation recharge hypothesis' [116], subsidence is placed against questions of water scarcity, particularly in urban parts that witness recurrent flooding. The belief holds that when it rains, the groundwater is replenished with precipitation, thus recharging depleted 'stocks' of groundwater. Arguably, increasing trends in subsidence in conjunction with the necessity for drilling deeper wells, at times reaching down to over 120 metres in the case of northern Jakarta, undermines this once pervasive myth which is being articulated far less in recent times. Yet, irrespective of whether groundwater levels are restored, soil compaction still remains mostly inelastic, as elastic compaction accounts for a small fraction of total compaction. Contrarily, during household interviews at *kampung*-level, residents believed that coastal spaces were less likely to be affected by subsidence than the hinterlands on higher elevations. It was also argued that the underlying cause of sinking land was due to the increased frequency and volume of rain-fed floods (*rob*, in Javanese), which in turn "made the soils heavy enough to sink in" (interview at Kamal Muara, March 2020).

In the cases of both Jakarta and Metro Manila, the importance of non-anthropogenic effects have hinged on the importance of auto-compaction as a result of tectonic shifting and the consolidation of alluvial soils [64, 82, 116]. Instead of excess carrying capacity that should be

monitored, the question of soil consistencies and compaction often took centre stage in policy debates (interview with Association of Urban Planners, Jakarta, February 2017). With repeated reference to problematic site selection for the establishment of a capital under the Dutch, 'sinking' is portrayed as inevitable (interview at Jakarta Municipality, Office for Coastal Protection, February 2017).

### 6.3. Individualization: Assigning blame for subsidence

The third perspective centres on the individualisation of LS. The relational blame game in Jakarta and Metro Manila involves questions of excess groundwater pumping that have conventionally been blamed on both poorer informal and often migrant settlements using artisanal pumps, and groupings of factories with highly efficient deep well systems. Moreover, differing structures of property ownership and tenure makes the burden of responsibility less certain between residential land owners, the state, property developers and industrial complexes. Consider for example the Singaporean Housing Development Board´s 99-year apartment leases, or Jakarta´s Communal Certificates of Entitlement for urban *kampung* residents. In such situations, accounting for the potentialities of subsidence bears implications on building costs and the devaluation of land.

Urban coastlines particularly in the global south exist as socially unequal spaces, insofar that they are often occupied by both the poor and the wealthy, at times in relatively close proximity. Metaphors such as "gating subsidence" were referred to in affluent spaces of northern Jakarta such as Waduk Pluit. Often, high-value residential land was raised every few years within their walled-in condominium blocks. Not too far from these affluent coastal edges, those residing in informal settlements often referred to their own small-scale land reclamation efforts (*nimbun)* as both a result of, and a solution for subsiding land.

## 7. Conclusion

In recent decades, the 'sinking city' has emerged as a powerful apocalyptic symbol for communicating the hazards and challenges posed by sea level change. In particular, low-lying megacities such as Jakarta and Metro Manila have historically placed more emphasis on mitigating and adapting to hazard-related extreme weather events. The complexities of anthropogenic LS, particularly along densely built urban coastlines, have been gaining greater public awareness and policy traction. Based on research into land subsidence as a geologic and socio-political phenomenon, the current article aimed to assess core challenges in recognising and addressing this phenomenon as an understudied governance issue in urban Southeast Asia. In sum, the paper calls for a mix of context-specific policy responses that not only consider the inequities of groundwater overexploitation, urban water supply and demand. The study underlines the importance of considering the broader politics of contemporary coastal transformations, particularly against the backdrop of urban building densification, land use policy, tenurial insecurity, forced relocation and land reclamation. Furthermore, these diverse dynamics increasingly interact in relation to upscale coastal real estate development.

Policy-level discussions on LS have often hinged on the multi-sectoral complexity of the issue. Complex because the localized dynamics of LS remain a political conundrum, given the urgency of arresting faster sinking land as opposed to regionally rising sea levels. Furthermore, LS remains a phenomenon that affects almost every domain of everyday urban life, from land use practices to urban water demand, disaster risk reduction politics to solutions for climate change mitigation and adaptation. At first glance, subsidence does not appear as a direct climate-related hazard. Yet its conflation as a distinctly urban problem affecting low-lying cities at risk to sea level change particularly in the global south, renders this framing problematic,

from not only from a multi-sectoral policy angle, but also from an interdisciplinary perspective spanning the geological, hydrological and the marine.

How far can the governance of LS be read as a classic 'wicked' or even as a super-wicked problem? At first glance, land subsidence comprises a number of characteristics that render it a textbook wicked problem. Its wickedness is derived in part from the need for a clear, overarching issue definition and a concerted multi-sectoral policy strategy in order to address it. Arguably, this wickedness assumes a double bind through two pervasive policy challenges: a) the absence and fragmented nature of policy measures themselves in arresting the multi-pronged root causes of subsidence; and b) the unfolding of policy being an embattled terrain where vested power interests play out. Its super-wickedness becomes apparent not only due to the sheer urgency of rapid LS rates, but also due to the fact that no central institutional entity exists in Jakarta and Metro Manila in integrating and enforcing anti-subsidence policy action, spanning both adaptation as well as mitigation. One question begs to be asked: what is lost (and gained) by *subsuming* LS under the mantle of international or regional climate change policy, particularly if solutions to such a localized phenomenon ought to remain localized and context-specific? Thus, another dimension of its wickedness owes as much to its scalar politics and thematic conflation.

On the other hand, LS exists as a coherent scientifically measurable reality, and is clearly enhanced by a set of anthropogenic factors. Yet policy action remains further clouded by the combination of natural processes characteristic of multi-hazard contexts. Political attention is regularly drawn elsewhere and the urgency for immediate action ignored due to the frequency of trigger events such as monsoonal flooding or tectonic activity. The enforcement of regulations such as reliable monitoring or an equitable licensing system would contribute to better coordinated mitigation and adaptation policies, yet cannot singularly arrest subsidence rates. Instead, the paper argues for cross-disciplinary, well-coordinated and scientifically informed policy action, further augmented by the lessons of neighbouring regional cities, but adapted to and translated into respective local societal contexts.

Furthermore, the socio-ecological complexity of LS calls for the strengthening of interdisciplinary scholarly engagement with its presence as a distinct coastal urban phenomenon in itself. In advancing these debates, several nascent themes have been identified: a) the framing of LS as both a hidden and perceived socio-political reality in everyday life, particularly in terms of whose forms of knowledge count and how; b) critical analyses on the spatial and temporal observation of the effects and causes of LS spanning traditional and emerging geodetic technologies, combined by grassroots monitoring; and c) the contested politics of redressing localised subsidence patterns across multiple local, regional and international scales in ways that render complexity in translating global IPCC and other dominant projections of mean SLR into coherent municipal action. In sum, because inclusive policymaking and regulatory action are both time and resource-intensive, substantial political will at both national and municipal levels becomes a prerequisite. Yet, the urgency of governing subsidence as a multi-sectoral, socio-ecological issue in its own right will need to gain greater political momentum in the years to come.

## Supporting information

**S1 File.**
(XLSX)

## Acknowledgments

We gratefully acknowledge Lucas Barning for the design input, together with Cora Bolong, Wafa Fauzia, Irene S. Fitrinitia, Arif Gandapurnama, Fajar Hatorangan, Nala Hutasoit, Devita

Rahmadani, Saemo Kamoyaki Rumengan, and Soly Santoso for their fieldwork assistance in Manila and Jakarta between February 2017 and March 2020. We also extend our thanks to Michael Flitner (University of Bremen) and Heri Andreas (ITB Bandung, Indonesia), Bayu Triyogo Widyantoro (BIG, Indonesia). Thanawat Bremard (G-EAU, IRD, ABIES Doctoral School, France), and Irina Rafliana (LIPI, Indonesia) for discussions about subsidence in Jakarta and the Southeast Asian region. We also highly appreciate the provision of tide gauge data for Jakarta from Badan Informasi Geospasial (BIG) in Cibinong, Indonesia.

## Author Contributions

**Conceptualization:** Rapti Siriwardane-de Zoysa, Tilo Schöne, Johannes Herbeck, Julia Illigner, Emma Porio.

**Data curation:** Rapti Siriwardane-de Zoysa, Tilo Schöne, Johannes Herbeck, Alessio Rovere.

**Formal analysis:** Rapti Siriwardane-de Zoysa, Johannes Herbeck, Julia Illigner, Mahmud Haghighi.

**Funding acquisition:** Tilo Schöne, Alessio Rovere, Anna-Katharina Hornidge.

**Investigation:** Rapti Siriwardane-de Zoysa, Tilo Schöne.

**Methodology:** Rapti Siriwardane-de Zoysa, Tilo Schöne, Johannes Herbeck, Julia Illigner, Mahmud Haghighi.

**Resources:** Tilo Schöne, Anna-Katharina Hornidge.

**Software:** Tilo Schöne, Mahmud Haghighi.

**Supervision:** Tilo Schöne.

**Visualization:** Rapti Siriwardane-de Zoysa, Tilo Schöne, Julia Illigner, Mahmud Haghighi.

**Writing – original draft:** Rapti Siriwardane-de Zoysa, Tilo Schöne, Johannes Herbeck, Julia Illigner, Mahmud Haghighi, Hendricus Simarmata, Anna-Katharina Hornidge.

**Writing – review & editing:** Rapti Siriwardane-de Zoysa, Tilo Schöne, Johannes Herbeck, Alessio Rovere, Anna-Katharina Hornidge.

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
