## [Decision Letter · Decision Letter 0]

1 Jun 2020

PONE-D-20-03843

The Wickedness of Governing Land Subsidence: Policy Perspectives from Urban Southeast Asia

PLOS ONE

Dear Dr. Siriwardane-de Zoysa,

Thank you for submitting your manuscript to PLOS ONE. After careful consideration, we feel that it has merit but does not fully meet PLOS ONE’s publication criteria as it currently stands. Therefore, we invite you to submit a revised version of the manuscript that addresses the points raised during the review process. In particular, the paper needs more work in the methods section, and the language could be plainer in some paragraphs, so the paper can reach a wider audience. 

We look forward to receiving your revised manuscript.

Kind regards,

Vanesa Magar, Ph.D.

Academic Editor

PLOS ONE

Journal Requirements:

2. Please amend your title to adhere to PLOS's submission guidelines: https://journals.plos.org/plosone/s/submission-guidelines#loc-title In this case we have concerns that the title may not be descriptive and specific enough, for instance the specific meaning intended by wickedness may not be clear to readers."

3. Please provide additional details regarding participant consent. In the ethics statement in the Methods and online submission information, please ensure that you have specified (1) whether consent was informed and (2) what type you obtained (for instance, written or verbal, and if verbal, how it was documented and witnessed). If your study included minors, state whether you obtained consent from parents or guardians. If the need for consent was waived by the ethics committee, please include this information.”

4. Please ensure that you include a title page within your main document. We do appreciate that you have a title page document uploaded as a separate file, however, as per our author guidelines (http://journals.plos.org/plosone/s/submission-guidelines#loc-title-page) we do require this to be part of the manuscript file itself and not uploaded separately.

6. We note that you have indicated that data from this study are available upon request. PLOS only allows data to be available upon request if there are legal or ethical restrictions on sharing data publicly. For more information on unacceptable data access restrictions, please see http://journals.plos.org/plosone/s/data-availability#loc-unacceptable-data-access-restrictions.

7. Please remove your figures from within your manuscript file, leaving only the individual TIFF/EPS image files, uploaded separately.  These will be automatically included in the reviewers’ PDF.

8. Please amend your manuscript to include your abstract after the title page.

9. We note that Figure 1 in your submission contain [map/satellite] images which may be copyrighted. All PLOS content is published under the Creative Commons Attribution License (CC BY 4.0), which means that the manuscript, images, and Supporting Information files will be freely available online, and any third party is permitted to access, download, copy, distribute, and use these materials in any way, even commercially, with proper attribution. For these reasons, we cannot publish previously copyrighted maps or satellite images created using proprietary data, such as Google software (Google Maps, Street View, and Earth). For more information, see our copyright guidelines: http://journals.plos.org/plosone/s/licenses-and-copyright.

10. Please include captions for your Supporting Information files at the end of your manuscript, and update any in-text citations to match accordingly. Please see our Supporting Information guidelines for more information: http://journals.plos.org/plosone/s/supporting-information.

Additional Editor Comments (if provided):

Reviewers' comments:

Reviewer's Responses to Questions

**Comments to the Author**

1. Is the manuscript technically sound, and do the data support the conclusions?

Reviewer #1: Yes

Reviewer #2: Yes

2. Has the statistical analysis been performed appropriately and rigorously? 

Reviewer #1: Yes

Reviewer #2: N/A

3. Have the authors made all data underlying the findings in their manuscript fully available?

Reviewer #1: Yes

Reviewer #2: No

4. Is the manuscript presented in an intelligible fashion and written in standard English?

Reviewer #1: Yes

Reviewer #2: Yes

5. Review Comments to the Author

Reviewer #1: In this passed year there are a bit improvement from the Government of Indonesia concerning the land subsidence issue. There is major project in respond to subsidence in Jakarta and others city along northern coast of Java Island. The Giant Sea Wall program is progressing a little bit. Now the reclamation is also winning by the law to continue. Maybe the author can improved a bit about this policy. It is just a minor addition

Reviewer #2: Dear Authors, Dear Editors,

I carefully read the manuscript PONE-D-20-03843. The paper combines InSAR-derived land subsidence measurements in three major cities of Southeast Asia with an analysis on how the problem is considered in local administrations. I am not familiar with social science papers, but I found the mixt approach between social sciences and geodesy VERY interesting: a rare but very valuable perspective on policies and local perceptions of the problem. I can recommend publications of the manuscript after a major editing work. Note: It would have been easier to review with line numbers on the PDF file.

Comments:

There are too many long sentences. In order to have impact on the widest audience possible (InSAR experts, hydrogeologists, policy makers, water managers, social scientists etc.), I strongly recommend to shorten/simply all sentences of more than 4 lines throughout the paper.

Section 2.2: Several technical details are missing in the InSAR process methods: how many images per site? Where is the reference point for each processing?

Page 8-9: Why does your InSAR measurement does not show subsidence? Can you explain why the subsidence rates have changed? It appears that everything that is detected over Singapore is within typical noise levels of InSAR processing, hence no recent subsidence.

Page 9 and part 2.2: InSAR measures Line-Of-Sight displacements, explain how the conversion to vertical displacements was done and what are the related assumptions (to insert in the method section).

‘Municipal freshwater supply substantially increased the likelihood of illegal dependence on

groundwater sources’ -> Reformulate.

Part 6.3: the reason why subsidence is irrecoverable is not only because groundwater needs cannot decreased, it is also and mainly because, even if the groundwater level get restored, compaction is mostly inelastic. Elastic compaction is usually a small fraction of the total compaction.

In the intro, to increase the international visibility of the paper, I would suggest listing/citing the main regions in the World that are undergoing land subsidence (apart from the one you study): Central Valley, Central Mexico, Venice, North China Plains:

https://www.nature.com/articles/srep02710

https://www.nature.com/articles/s41598-019-52371-7

https://doi.org/10.1016/j.jag.2015.12.002

https://www.mdpi.com/2072-4292/10/3/365

6. PLOS authors have the option to publish the peer review history of their article (what does this mean?). If published, this will include your full peer review and any attached files.

Reviewer #1: Yes: Heri Andreas

Reviewer #2: No

---

## [Author Response · Author response to Decision Letter 0]

10 Sep 2020

Response to Editors and Reviewers – The “wickedness” of governing land subsidence: Policy perspectives from urban Southeast Asia

Editorial concerns

1. We have checked the file templates and the following changes were made: a) to the titling in the manuscript, together with the appropriate line numbering, double spacing, and pagination (in keeping with the submission guidelines); b) the referencing has been changed to Vancouver style; c) all footnotes have been deleted 

2. We have placed wickedness in inverted commas in order to draw attention to its policy peculiarity. A conceptual description has been given in section 1.3 of the paper. 

3. Details regarding verbal participant consent have been provided in the ethics statement in the Methods and online submission information. No minors participated in the study. 

4. The title page has now been uploaded as a separate file. 

5. An ORCID ID has been created. 

6. In the cover letter we have detailed our ethical restrictions in sharing our in-depth interview transcripts. However recent interview survey material gleaned in March 2020 has been uploaded. 

7. Figures from the manuscript have been removed. Figs 1 and 2 have been uploaded in .tiff format. 

8. Abstract has been added after the title page. 

9. On copyrights pertaining to the satellite images: We used Copernicus and SRTM data in the figure. Copernicus data is copyright of ESA, but we can freely use them as long as we provide appropriate credit.

We have thus added the following statement to the figure caption:

"Contains Copernicus data 2018.”

i. More information on Sentinel-1 data credit in the following links:

https://sentinel.esa.int/web/sentinel/terms-conditions

https://scihub.copernicus.eu/twiki/pub/SciHubWebPortal/TermsConditions/TC_Sentinel_Data_31072014.pdf

SRTM data is public domain and can be used without any restriction:

https://catalog.data.gov/dataset/shuttle-radar-topography-mission-dted-level-1-3-arc-second-data-dted-1

We have also added the following SRTM publication in the references: 

Farr, T. G., and Mike K. "Shuttle Radar Topography Mission produces a wealth of data." Eos, Transactions American Geophysical Union 81.48 (2000): 583-585.

10. Where relevant, captions for supporting information files have been provided at the end of the manuscript with updates for the in-text citation. 

Concerns raised by the reviewers:

1. The last empirical section of the paper (Section 6) has been strengthened with recent fieldwork findings from Jakarta that were gleaned in March 2020. 

2. To Reviewer 1 (R1): With regard to the data availability statement, we have provided anonymised quantified data for a recent social survey conducted in March 2020. The interview transcripts themselves are confidential as anonymity was guaranteed to all participants as their narratives/discussions were taped. 

3. R1: A several paragraphs have been added on more recent policy development in Jakarta, including the Great Garuda megaproject. Please refer to the following lines: 569-575 and 650-671. We have touched upon Jakarta’s impending relocation plans elsewhere in the paper. However, for the sake of brevity, we have not delved into this in detail. 

4. Reviewer 2 (R2): All long-winded sentences have been shortened and redrafted for clarity and brevity. These sections have been highlighted in yellow or marked out in the manuscript as tracked changes. We have also checked for repetitions and deleted them. 

5. R2 (on geodesy data): 

- We have added a new table (Table 1) to the paper. The newer map has been added (see Fig 1), replacing the former. 

- The question regarding reference points for constraining our maps: 

To constrain the InSAR displacements, we used GNSS measurements in Jakarta and Manilla (Displayed by circles in Figure 1). In Jakarta, a GNSS station without significant displacement is used and in Manila, the vertical displacement from PIMO IGS permanent station is used to constrain the vertical displacement from InSAR. In Singapore, we did not use any external information and constrained the InSAR displacements by assuming the average of all InSAR observation points is zero. 

- With regard to the values in Singapore above the RMS:

The RMS of displacement rate in Singapore is as small as 2.5 mm/a. There are areas with displacement rates higher than the estimated RMS. Because we did not have any absolute GNSS point in this area, we constrain the displacement rates by assuming the displacement rates average zero. Such assumption might slightly bias the estimated displacement rates. Therefore, small-scale displacements within the range of a few mm/a should be interpreted very carefully. In several reclamation areas, as discussed in the manuscript, the displacement rates can be reliably interpreted as land subsidence.

- In order to explain the conversion from LOS to vertical: 

InSAR measures the displacement in satellite Line-of-Sight (LOS) direction which is defined by geometry of the sensor. Assuming the horizontal components of displacement are negligible, we use the incidence angle (θ) to convert the LOS measurement to vertical displacement V = LOS / cos(θ).

6. We have reformulated a number of ambiguous sentences that were highlighted, including part 6.3, and the point raised on inelastic compaction. We thank the Reviewer 2 for raising this point. 

7. We have added to the introduction other subsiding regions around the world, while citing the papers that were shared by R2.

---

## [Decision Letter · Decision Letter 1]

13 Nov 2020

PONE-D-20-03843R1

The ‘wickedness’ of governing land subsidence: Policy perspectives from urban Southeast Asia

PLOS ONE

Dear Dr. Siriwardane-de Zoysa,

Thank you for submitting your manuscript to PLOS ONE. After careful consideration, we feel that it has merit but does not fully meet PLOS ONE’s publication criteria as it currently stands. Therefore, we invite you to submit a revised version of the manuscript that addresses the minor points raised by the reviewer. 

We look forward to receiving your revised manuscript.

Kind regards,

Vanesa Magar, Ph.D.

Academic Editor

PLOS ONE

Reviewers' comments:

Reviewer's Responses to Questions

**Comments to the Author**

1. If the authors have adequately addressed your comments raised in a previous round of review and you feel that this manuscript is now acceptable for publication, you may indicate that here to bypass the “Comments to the Author” section, enter your conflict of interest statement in the “Confidential to Editor” section, and submit your "Accept" recommendation.

Reviewer #3: (No Response)

2. Is the manuscript technically sound, and do the data support the conclusions?

Reviewer #3: Yes

3. Has the statistical analysis been performed appropriately and rigorously? 

Reviewer #3: Yes

4. Have the authors made all data underlying the findings in their manuscript fully available?

Reviewer #3: Yes

5. Is the manuscript presented in an intelligible fashion and written in standard English?

Reviewer #3: Yes

6. Review Comments to the Author

Reviewer #3: Taking Jakarta, Manila and Singapore as case studies, the author defines land subsidence as a typical "wicked" policy issue and reveals a broader coastal governance pattern. But some revisions are needed before they can be published. The comments are as follows:

（1）How to validate subsidence results of sentinel-1? The article lacks the necessary validation results, such as graphs and tables. Please refer to ‘Time-Series Evolution Patterns of Land Subsidence in the Eastern Beijing Plain, China’.

（2）Is sea level rise only for data reference? Is there a relationship between subsidence and sea level rise? Is the condition of subsidence and sea level interaction used in the analysis? If so, please indicate.

（3）Fig.1 is not professional and lacks the basic elements of the map, such as the compass, the marking of rapid subsidence area and the coverage of sentinel-1. Please refer to the papers on land subsidence, such as‘Land Subsidence Response to Different Land Use Types and Water Resource Utilization in Beijing-Tianjin-Hebei, China’.

（4）The geographical location and boundaries of the study area are shown in the form of maps.

（5）In line 339, "the subordination levels" is not a descriptive term for a sinking scientific paper.

7. PLOS authors have the option to publish the peer review history of their article (what does this mean?). If published, this will include your full peer review and any attached files.

Reviewer #3: No

---

## [Author Response · Author response to Decision Letter 1]

10 Mar 2021

The original review statements and responses to the reviewers are furnished below:

(1) How to validate subsidence results of sentinel-1? The article lacks the necessary validation results, such as graphs and tables. Please refer to ‘Time-Series Evolution Patterns of Land Subsidence in the Eastern Beijing Plain, China’.

RESPONSE: 

The InSAR-Analysis and stacking of the time series follows the standard approach (Hooper et al., 2004 & Hooper et al., 2012). To ensure the geodetic consistency of the (relative) values of InSAR, we introduce the GNSS-derived vertical motion of reference points for constraining the individual InSAR maps, which is equivalently to the method using leveling points in the mentioned paper. The use of GNSS-points has the advantage that we can use a continuous time series of GNSS heights and the geodetic consistency between the different geodetic technologies.

(2）Is sea level rise only for data reference? Is there a relationship between subsidence and sea level rise? Is the condition of subsidence and sea level interaction used in the analysis? If so, please indicate.

RESPONSE:

The information about the sea level change in this area is derived from radar altimetry, which itself is a geocentric information, and from tide gauges, giving relative sea level information. For Jakarta the tide gauge analysis takes the GNSS-derived subsidence at the tide gauge into account. For Manila and Singapore no such information is available, thus we depend on the InSAR information. As we have constrained the InSAR stacking to a GNSS-reference point, we ensure the consistency between the different geodetic technologies.

(3）Fig.1 is not professional and lacks the basic elements of the map, such as the compass, the marking of rapid subsidence area and the coverage of sentinel-1. Please refer to the papers on land subsidence, such as ‘Land Subsidence Response to Different Land Use Types and Water Resource Utilization in Beijing-Tianjin-Hebei, China’.

RESPONSE:

PLOS One doesn’t require a North arrow (see https://journals.plos.org/plosone/article?id=10.1371/journal.pone.0237878, our maps are northwards oriented and have a grid annotation which clearly indicate the orientation and location. Additionally, since the paper is a study of social actions, the information about the swath orientation of Sentinel-1A doesn’t increase the information value.

(4) The geographical location and boundaries of the study area are shown in the form of maps.

RESPONSE:

Thank you for this useful suggestion. We have reworked the map as suggested with insets for each city.

(5) In line 339, "the subordination levels" is not a descriptive term for a sinking scientific paper.

RESPONSE:

“Subsidence rate “(not “subordination” was incidentally named level). This minor error has been corrected in the manuscript.

---

## [Editor Report · Decision Letter 2]

5 Apr 2021

The ‘wickedness’ of governing land subsidence: Policy perspectives from urban Southeast Asia

PONE-D-20-03843R2

Dear Dr. Siriwardane-de Zoysa,

We’re pleased to inform you that your manuscript has been judged scientifically suitable for publication and will be formally accepted for publication once it meets all outstanding technical requirements.

Kind regards,

Vanesa Magar, Ph.D.

Academic Editor

PLOS ONE

Additional Editor Comments (optional):

The authors have replied to the first reviewer's concerns in the response letter, not in the text. It would have been useful for other readers to clarify this information in the paper itself as well. However, they have addressed some other minor issues and therefore I recommend the paper for publication. 
---

## [Editor Report · Acceptance letter]

17 May 2021

PONE-D-20-03843R2 

The ‘wickedness’ of governing land subsidence: Policy perspectives from urban Southeast Asia 

Dear Dr. Siriwardane-de Zoysa:

I'm pleased to inform you that your manuscript has been deemed suitable for publication in PLOS ONE. Congratulations! Your manuscript is now with our production department. 

Kind regards, 

on behalf of

Dr. Vanesa Magar 

Academic Editor

PLOS ONE